# Probiotics in Wound Healing

**DOI:** 10.3390/ijms25115723

**Published:** 2024-05-24

**Authors:** Valentina Alexandra Bădăluță, Carmen Curuțiu, Lia Mara Dițu, Alina Maria Holban, Veronica Lazăr

**Affiliations:** 1Department of Microbiology, Faculty of Biology, University of Bucharest, 030018 București, Romania; badaluta.valentina@s.bio.unibuc.ro (V.A.B.); carmen.curutiu@bio.unibuc.ro (C.C.); lia-mara.ditu@bio.unibuc.ro (L.M.D.); veronica.lazar@bio.unibuc.ro (V.L.); 2Research Institute of the University of Bucharest—ICUB, University of Bucharest, 050657 Bucharest, Romania

**Keywords:** wound healing, probiotic formulations, infection control, wound dressings

## Abstract

Wound infections caused by opportunistic bacteria promote persistent infection and represent the main cause of delayed healing. Probiotics are acknowledged for their beneficial effects on the human body and could be utilized in the management of various diseases. They also possess the capacity to accelerate wound healing, due to their remarkable anti-pathogenic, antibiofilm, and immunomodulatory effects. Oral and topical probiotic formulations have shown promising openings in the field of dermatology, and there are various in vitro and in vivo models focusing on their healing mechanisms. Wound dressings embedded with prebiotics and probiotics are now prime candidates for designing wound healing therapeutic approaches to combat infections and to promote the healing process. The aim of this review is to conduct an extensive scientific literature review regarding the efficacy of oral and topical probiotics in wound management, as well as the potential of wound dressing embedding pre- and probiotics in stimulating the wound healing process.

## 1. Introduction

Wound healing involves intricated mechanisms, highlighting the multifaceted nature of this process. Wound repair is classically simplified into four main phases: hemostasis, inflammation, proliferation, and dermal remodeling. These phases result in both architectural and physiological restoration of the skin following damage, ensuring it can act as a primary defense barrier [1,2,3].

The initial phase of wound repair, hemostasis, plays a critical role in stopping bleeding and initiating the cascade of events that leads to tissue repair. Hemostasis involves several interconnected processes, including vasoconstriction, platelet activation, and coagulation, which work together to form a stable blood clot at the site of injury. Endothelial cells lining the blood vessels constrict in response to injury, reducing blood flow to the affected area. Concurrently, platelets adhere to the exposed extracellular matrix and undergo activation, releasing factors that promote further platelet recruitment and aggregation [3,4]. This phase is usually very susceptible to infection, since microbial cells could easily penetrate the damaged skin barrier and invade the surrounding tissues.

The inflammatory phase initiates the wound healing cascade, characterized by the influx of immune cells, such as neutrophils and macrophages, to the wound site. These cells release cytokines and growth factors that orchestrate tissue repair and clearance of debris and pathogens. Studies have elucidated the role of various signaling pathways, including the NF-κB (Nuclear Factor kappa-light-chain-enhancer of activated B cells) and MAPK (Mitogen-Activated Protein Kinase) pathways, in regulating inflammatory responses and promoting tissue regeneration [5]. During the proliferative phase, fibroblasts migrate into the wound bed and deposit extracellular matrix components, such as collagen and fibronectin, to form granulation tissue. Recent research has focused on understanding the molecular mechanisms underlying fibroblast activation and migration, as well as the role of growth factors, such as TGF-β (Transforming Growth Factor-β) and PDGF (Platelet-Derived Growth Factor), in promoting cell proliferation and angiogenesis [6,7]. Additionally, studies have highlighted the importance of the endothelial-to-mesenchymal transition in contributing to the pool of activated fibroblasts during wound healing [8]. Such inflammatory factors are also involved in infection control, as the microbial cells present in the lesion area can be detected and neutralized by host immune effectors. However, numerous wound pathogens evade such host immune responses and initiate wound infections, which could be acute and subsequently cleared, or they may establish persistent and chronic infections that are almost impossible to eradicate by the host’s immunity or classical antibiotherapy.

During the proliferation phase of wound repair, the focus shifts to rebuilding tissue and filling the wound gap. One key aspect of the proliferation phase is the migration and proliferation of various cell types essential for tissue repair. Fibroblasts are pivotal in this process, as they migrate into the wound bed and produce extracellular matrix components, such as collagen, fibronectin, and proteoglycans [9]. Recent studies have investigated the signaling pathways regulating fibroblast activation and migration, including TGF-β, PDGF and FGF (fibroblast growth factor), which stimulate cell proliferation, and ECM (extracellular matrix) synthesis [10,11,12]. Endothelial cells also play a crucial role during the proliferation phase by promoting angiogenesis, the formation of new blood vessels, to restore blood supply to the wound site. The VEGF (vascular endothelial growth factor) and angiopoietins are key regulators of angiogenesis, orchestrating endothelial cell migration, proliferation, and tube formation [13,14]. Moreover, epithelial cells at the wound edges undergo migration and proliferation to resurface the wound and restore the epithelial barrier.

The remodeling phase involves the maturation and remodeling of the newly formed tissue to restore its structural integrity. Collagen fibers undergo cross-linking and realignment, while the excess extracellular matrix is degraded by matrix metalloproteinases (MMPs). Recent research has investigated the dynamic regulation of MMP activity and collagen turnover during the remodeling phase [15], as well as the role of mechanical forces and tissue tension in modulating scar formation and wound contraction [16]. Wound contraction involves the gradual reduction in wound size, mediated by myofibroblasts [17]. Additionally, research has identified factors such as TGF-β and mechanical cues, which regulate myofibroblast activation and function during wound contraction. The cellular and molecular mechanisms underlying scar maturation include the alignment and reorganization of collagen fibers, as well as the modulation of scar pigmentation and vascularity [18].

The presence of pathogenic microorganisms interferes with such proliferation and remodeling processes in the wound area and is often responsible for delayed wound healing and improper tissue repair, leading to long-term invalidity and patient discomfort.

Many of the classical wound pathogens are antibiotic-resistant microorganisms, which represent a serious global concern, as they increase the risk of developing healthcare-associated infections (HAI) in hospital and surgical units.

Moreover, tolerant microorganisms causing biofilm infections are another huge threat in the development of chronic and difficult-to-treat wounds. In this context, more efficient approaches to fight such microbial challenges are necessary, representing a global priority.

Current alternatives to conventional antibiotics in combating wound infections include bacteriophage therapy, antimicrobial peptides, oral and topical probiotics, antibody therapy, and antibacterial nanomaterials, etc. [19].

It is well known that the microbiome can affect the normal physiology of several host organs and systems, while a disturbance in the microbiota community structure, or a failure of the host’s responsive mechanisms to the microbiome, has been implicated in a variety of disease etiologies. For example, the oral microbiome impacts the healthy oral tissue homeostasis, the therapeutic options of treatment with live bacteria for diseased states within the oral cavity, and it may influence the healing of oral wounds resulting from oroantral and oronasal fistulas. In addition, recent studies are investigating the oral microbiome in relation to its influence on body organs distant from the alimentary canal, on systemic health, and disease. Therefore, the utilization of live microorganisms obtained from normal microbiota, or those with probiotic potential, to treat different chronic diseases is a hot topic that has gained significant attention among researchers [20].

According to the Scientific Association of Probiotics and Prebiotics, “probiotics are live microorganisms which provide health benefits when they are consumed in an adequate manner” [21].

The concept that probiotics promote wound healing has been studied for a number of years. Researchers have elucidated some anti-pathogenic mechanisms of probiotics, which are useful for improving the healing process, i.e., inhibition of pathogen attachment, competition for adherence sites, nutrients, and other vital resources, production of anti-pathogenic molecules, antagonistic activity [22], as well as stimulation of epithelial barrier function and modulation of the immune response [23].

Many clinical trials, in vitro and in vivo experiments conducted on various animal models, demonstrated the ability of probiotic formulations to improve tissue repair and to modulate the host immune system.

This paper aims to review recently published studies focused on the anti-pathogenic activity of probiotics against microorganisms implicated in wound infections and on their application in wound healing, especially the benefits of orally ingested probiotic supplements and topical formulas.

## 2. Types of Probiotics in Wound Healing

Probiotic therapy has a promising role in promoting the healing of dermatological conditions, including chronic wounds, surgical wounds, diabetic foot ulcers (DFU), venous or arterial ulcers, oral wounds, atopic dermatitis, pressure sores, acne vulgaris, hidradenitis supurativa, etc.

The most commonly identified microorganisms causing dermatologic conditions are represented by biofilm-forming bacteria such as *Pseudomonas aeruginosa*, *Escherichia coli*, *Peptostreptococcus* sp., *Staphylococcus aureus*, *Enterobacter* sp., *Klebsiella pneumoniae*, *Acinetobacter baumanii*, *Enterococcus faecalis*, etc. [24].

Probiotics could improve the healing process by inhibiting the growth of planktonic Gram-negative and Gram-positive bacteria and limiting their biofilm development by releasing molecules that interfere with the quorum-sensing (QS) system of bacteria and by blocking the adhesion process to the epithelial tissue [25].

Over the past decade, the interest in the topical application of probiotic products and oral probiotic formulations has significantly expanded.

The therapeutical applications of probiotics and their derivates, including culture supernatants and mixed probiotics, have been investigated in vivo, on animal models and in humans, but also in vitro, in tissue cultures. These studies have demonstrated their beneficial wound-healing effects and their ability to reduce the colonization of pathogens through competition for substrate-adhesion sites, nutrients, and growth factors. Probiotics also interference with the QS signaling system by producing lactic acid, which decreases the pH of the local environment, and other anti-pathogenic molecules, such as hydrogen peroxide, reuterin, and bacteriocins. These substances are able to disrupt the most common chronic wound microbial pathogens or to inhibit their virulence [26,27]. Additionally, another antimicrobial mechanism is through the regulation of antimicrobial peptide (AMPs) production by the host’s epithelial cells, adipocytes, and mast cells, which modulate skin integrity, reduce inflammation, and prevent adhesion and biofilm development [28].

The most predominant species of probiotics studied as new therapeutical strategies for their capacity to promote wound healing, making them ideal healing candidates, include lactic acid bacteria (LAB) such as *Lactobacillus plantarum*, *L. acidophillus*, *L. fermentum L. paracasei*, *L. rhamnosus*, *L. bulgaricus*, *L. reuteri*, *L. delbreuckii*, *L. salivarus,* and *Bifidobacterium lactis* [25,29]. LAB are Gram-positive, microaerophilic, and non-sporulating microorganisms with health benefits for the host, especially regarding dermal health, by improving the regulation of the skin and mucosa immune system and by maintaining cutaneous homeostasis [30]. In addition, studies showed that species from the genera *Bacillus*, *Enterococcus*, *Streptococcus,* or yeasts such as *Saccharomyces*, could be used as probiotics with efficient wound healing properties [26,31].

### 2.1. Oral Probiotics

Oral administration of probiotic formulations sets off a cascade of systemic effects that can indirectly impact wound healing, such as improving the absorption of minerals, vitamins, and cofactors implicated in the regulation of skin healing, reducing inflammation, regulating the host’s immune system, and modulating the microbiota composition. These effects further promote a balanced health condition in the host.

Oral administration of probiotics has been analyzed to evaluate wound healing in thermal burns, surgical lesions, diabetic foot ulcers, oral mucosa injuries, and episiotomy wounds [32].

Studies have shown that *Lactobacillus casei* oral supplementation is effective in healing episiotomy wounds. A triple-blind, randomized clinical trial was performed on 74 primiparous women who delivered in Iran. Participants with mediolateral episiotomy (incision length equal to and less than 5 cm) were randomly assigned to the probiotic and placebo groups, and the probiotic group received *L. casei* 431 at a dose of 1.5 × 10^9^ colony-forming unit (CFU)/capsule once a day, from the day after birth to 14 days postpartum. Wound healing, as a primary outcome, was measured by redness, edema, ecchymosis, discharge, and approximation; pain, as a secondary outcome, was measured by the Visual Analogue Scale before discharge and at 5 ± 1 and 15 ± 1 days after birth. As very good results were obtained in the wound healing study, the authors also suggested evaluating the effect of topical use *of L. casei* on episiotomy repair and pain in further studies [33].

In a recent clinical trial conducted by Wälivaara and colleagues, the potential therapeutic effects of probiotic lozenges on oral wounds were investigated. *L. reuteri* (DSM 17938 and ATCC PTA 5289) has been found to possess healing properties due to its capacity to limit local bacterial growth in oral mucosa injuries. The study was performed on 64 patients with a history of pericoronitis after surgical removal of mandibular third molars. They were randomly allocated into two groups: patients treated with lozenges which contained at least 2 × 10^8^ live bacteria, three times a day for two weeks; and a placebo group, who received identical lozenges, but without live bacteria. After the surgical extraction, most patients reported less swelling and alleviated symptoms, with no significant differences between the placebo and the probiotic groups [34].

Another clinical trial revealed the beneficial effects of oral probiotics on diabetic foot ulcers. A randomized double-blind clinical trial showed that oral probiotics, including *L. fermentum*, *L. acidophilus*, *L. casei,* and *Bifidobacterium bifidum,* incorporated in capsules administered daily on a period of 12 weeks among patients with grade-three diabetic foot ulcer, reduced the ulcer length and significantly improved the biochemical parameters of healing, including total cholesterol, plasma nitric oxide, and malonialdehyde levels [35].

Moreover, a randomized, controlled, prospective clinical trial, involving 90 male pediatric patients under two years of age with surgical intervention for hypospadias, aimed to investigate the effects of probiotics associated with antibiotic therapy. The study was performed on three groups of patients: the first group received antibiotics and oral probiotics in the form of drops containing *L. rhamnosus* GG ATCC 53103, once a day, for 4–16 days, depending on the duration of the treatment with antibiotics; the second group was treated only with antibiotics, while the placebo group received drops of a glucose solution at 5%. The study provided evidence for oral probiotics as a preventive measure for antibiotic-associated diarrhea and related clinical post-operative complications in pediatric patients. A notable decrease in the frequency of dressing change, the duration of antibiotic-associated diarrhea, and the incidence of postoperative wound complications, including the dehiscence of the reconstructed skin or fistula, were observed with the probiotic group. This outcome confirmed that oral probiotics could be used as a complementary therapy for the management of complications which may occur after surgical intervention for hypospadias repair [36]. Although the results were statistically significant, the study has some drawbacks, for example, the lack of a probiotic-only control group. The inclusion of a control group would provide important information on whether the tested probiotics may promote hypospadias repair on their own, or if they can only improve the effects of antibiotherapy and/or alleviate antibiotic-related symptoms such as diarrhea and reconstructed skin dehiscence in patients.

It seems that the administration of probiotics to burn patients could improve the inflammatory parameters, according to a study performed by Putra et al. [37]. A double-blind, randomized close-label study was conducted on 23 critically burn patients, separated into two groups, both of which were treated once a day, for 14 days, with monostrain probiotics that contained 10^7^ CFU (colony forming units) of either *L. acidophilus* or *Bifidobacterium longum*; or multistrain probiotics, including four strains of *Lactobacillus* (*L. rhamnosus*, *L. delbrueckii subsp. Bulgaricus*, *L. acidophilus* and *L. casei*), one strain of *Streptococcus* (*Streptococcus salivarus subsp. Thermophilus*), and two strains of *Bifidobacterium* (*B. breve* and *B. longum*), in form of capsules, administered orally. The authors demonstrated the capacity of both mono- and multistrain probiotics administered on oral route to significantly enhance the immune response by decreasing the neutrophil and leucocyte levels. Moreover, probiotics proved to limit the microbial growth and biofilm development of some recognized burn pathogens, while indirectly improving wound healing [38].

Table 1 bellow clusters the most recent studies aiming to reveal the impact of orally administered probiotics as therapeutic alternatives or adjuvants in wound healing.

### 2.2. Topical Probiotics

Topical applications of probiotics have been proposed as alternatives to antibiotics since 1912, as a method to treat cutaneous infections [40].

Probiotic formulation treatment applied topically is an attractive option for improving wound healing efficacy through promoting cutaneous tissue repair and reducing bacterial growth and development.

Topical probiotics have proven their efficiency in burns, skin ulcers, or surgical lesions. Several investigations were performed on animal models, including rats, pigs, or mice, wherein the wounds were inoculated with recognized wound-related microorganisms, including *S. aureus*, *P. aeruginosa*, *A. baumanii*, and *E. coli*, as well as with topical probiotics, metabolites, or supernatants [30,41,42,43,44].

Cutaneous probiotic strains have shown several beneficial properties, by promoting a continuous healing process through increasing the ceramide synthesis in keratinocytes, which are associated with a healthy skin–lipid barrier [45]. Additionally, the production of acids and antimicrobial molecules includes antimicrobial effects against pathogens implicated frequently in wound infections. Topical application of probiotics could induce different growth-reducing capacities towards relevant wound pathogens, as demonstrated in a novel in vivo-like human plasma biofilm model (hpBIOM). This study showed that probiotics such as *B. lactis* and *S. cerevisiae* possess slight bacteria-reducing properties, while the survival of the opportunistic yeast *C. albicans* is not affected at all by their presence. The most significant antimicrobial activity was detected, however, in all the evaluated wound pathogens (namely, *P. aeruginosa*, *S. aureus*, *S. epidermidis*, *E. faecium,* and *C. albicans*) after the treatment with a probiotic strain of *L. plantarum* [38].

Even though the mechanisms of topical probiotic formulations have not yet been clearly revealed, Table 2 below lists relevant studies performed both in vitro and in vivo, discussing their beneficial properties on the wound healing process.

## 3. Mechanisms of Probiotics in Wound Healing

Despite their proved efficiency in the therapy of wound infections, the intimate mechanisms of probiotics in wound healing are not fully understood.

Recent research was focused on their clinical application in chronic wound infections by improving vascularization and epithelization, modulating inflammatory process and immune response, promoting tissue repair, and limiting the colonization of pathogens [57,58].

Wound healing is a physiological process that occurs in an organism as a result of an injury. Many types of cells are involved in this process, and more steps are taken for tissue repair: (i) a homeostasis/inflammatory phase that limits the damage by closing the wound and eliminating cellular debris and bacteria; (ii) one proliferative phase for re-epithelialization and neovascularization; and finally; (iii) the maturation and remodeling phase. If for some reason the wound fails to heal in 4 to 6 weeks, it means that patients are dealing with chronic wounds [9]. Bacterial colonization and/or bacterial biofilm development at the wound site or the surrounding tissue are factors that favor and maintain chronic wounds. For successful healing, an antimicrobial approach is recommended. Due to the alarming increase in antibiotic resistance, probiotics represent a possible solution due not only to the fact that they possess some anti-pathogenic mechanisms but also to the fact that they promote tissue repair.

The presence of a high quantity of microorganisms in a wound prevents healing. The more microorganisms a wound has, the harder it is to heal. Moreover, when the tissue is damaged, the microbiota at that level could change, and some pathogenic species could multiply in excess, aggravating the damage. As is very well known, one of the essential properties of probiotics is their ability to compete with pathogenic microorganisms for nutrients and adhesion sites on host mucosa. Their species-specific antagonistic activity inhibits pathogenic species from colonizing that niche. Additionally, their antimicrobial activity is mediated by the production of lactic acid, resulting in the acidification of the matrix, but some strains are able to also produce bacteriocins [59]. Inflammation also occurs at this stage. Probiotics usually act as immunomodulatory triggers that sustain an influx of inflammatory cells at the wound site, including macrophages and polymorphonuclear leukocytes (PMN). They produce exopolysaccharides that have immunostimulatory activity and also stimulate the production of cytokines and chemokines. In both situations, the results consist in an afflux of neutrophils and macrophages to the site of infection. The phagocytosis process is also stimulated, leading to a decrease in the wound bacterial burden by reducing pathogens growth and attachment. In addition, inflammation is increased, a fact that accelerates the re-epithelialization process. Probiotics also exert their antimicrobial activity by regulating the production of antimicrobial peptides (AMPs) by the skin cells, molecules that modulate the skin microbiota composition and improve skin integrity [28].

Probiotics could also impact, at least indirectly, the proliferative phase, stimulating the re-epithelialization and neovascularization of the damaged tissue. The presence of probiotics leads to the stimulation of the pro-inflammatory M1 macrophage phenotype, followed by the immediate switch to M2, an anti-inflammatory response that promotes angiogenesis and epithelization [60]. If initially a pro-inflammatory response is necessary for microbial sterilization and the removal of foreign compounds, in the second phase of the healing process, a regression of neutrophils and macrophages is necessary to allow the scar to heal. The epithelialization phase is characterized by organized collagen deposition, with replacement of type III collagen with type I collagen, appropriate for the composition of the healthy dermis (with higher tissue strength). Neovascularization is necessary to ensure an adequate nutrient supply, immune cells, and oxygen for the proper epithelialization process [61].

Many studies using animal models have already demonstrated these in vivo effects of probiotics. It was observed that wounds treated with kefir had increased collagen levels and capillary vessels, as well as an earlier normal tissue reconstruction. Lactobacilli injected in acute wounds determined the proliferation of newly formed blood vessels. Nisin, clausin, and amyloliquecidin were also proved to be implicated in neovascularization and cell migration, as well as in the development of a thick epithelial layer [58]. Another study also described the capacity of some lactobacilli strains, such as *L. rhamnosus* and *L. plantarum*, to stimulate angiogenesis by up-regulating VEGF and/or EGF (Epidermal Growth Factor) expression. Additionally, other strains, such as *Saccharomyces boulardii*, were found to inhibit blood vessel growth induced by VEGF [61]. In another study on diabetic rats, orally administered probiotics resulted in a reduced wound area compared to the control group, due to an increase in type I collagen deposition and increased neovessel formation [62].

Remodeling is the final phase of the wound healing process. During this stage, the granulation tissue matures and becomes a scar with tensile strength, but also with flexibility. Matrix metalloproteases break down collagen III and replace it with collagen I, which further reorganizes into parallel fibrils, forming a scar with low cellularity. The *L. plantarum* and *Streptococcus thermophilus* probiotic strains were found to initiate earlier collagen III synthesis and deposition through TGF-β and then replace it with type I collagen. In addition, lactobacilli proved to accelerate keratinocyte migration and proliferation [63].

Figure 1 bellow summarizes the main probiotic mechanisms which could be associated with wound healing.

A recent study revealed that *Lactiplantibacillus plantarum*, *L. casei*, and *Lacticaseibacillus rhamnosus* decrease inflammatory cytokines and TNF- α and increase the production of anti-inflammatory cytokines in various studied murine models. Moreover, following the infection of mice with certain microbial strains, a reduced level of CRP (C-Reactive Protein) was observed, which strongly indicates that these probiotic strains could mitigate the inflammatory response. Moreover, it was observed that the wound area is reduced, and the healing process is faster when applying different formulations containing these strains [64].

Although *Enterococcus faecalis* is one of the most frequently isolated bacterial species in wounds, being capable to modulate host immunity and promote persistent infection and delayed healing, some Enterococcus strains have also been tested as probiotics. For example, Tanno et al. (2021) demonstrates that heat-killed *E. faecalis* KH2 stimulated TNF-α, interleukin-6, (TGF)-β1, and VEGF production and accelerated re-epithelialization and the formation of granulation tissue [65].

In a study that investigates potential *S. boulardii* effects on rats with *S. aureus*-induced skin infection, the healing potential of topically applied *S. boulardii* was demonstrated. As a possible mechanism for the obtained healing capacity of the studied yeast, a competitive exclusion of the colonizing pathogenic bacteria and the enhancing of antimicrobial peptide gene expression have been proposed [66].

*S. thermophilus* is another tested strain with probiotic potential for use in wound healing, due to its anti-pathogenic properties, revealed against numerous opportunistic bacteria involved in such infections. The experimental results on a mouse model showed that the inflammation in wounds treated with *S. thermophilus* was less intense, as compared to the control groups. If the inflammatory phase ends earlier, the proliferative phase will start in a shorter time; therefore, the entire wound healing process takes less time. Both the microscopic and macroscopic results showed that the wound healing rate was significantly higher in the experimental group compared to the control [67].

Bacteriocins are antimicrobial peptides produced by bacteria that eliminate other bacteria, offering the producing strain a competitive advantage for nutrients or habitats. Because their mechanisms of action are different from those of antibiotics, they could be effective against both antibiotic-sensitive and -resistant bacteria. Nisin is a lantibiotic, an antibacterial peptide produced by the Gram-positive bacteria of the *Lactococcus* and *Streptococcus* species. Recent findings suggested that nisin, along with kanamycin, helped reduce cytokine production, control local inflammation, and accelerate wound healing in rats. Nisin Z also modulated innate immune responses by inducing chemokine synthesis and suppressing LPS-induced proinflammatory cytokines. In addition, nisin can also be used in the design of improved wound dressings. Nisin A has been shown to have immunomodulatory effects and improve the wound healing process in human keratinocytes, human umbilical vein in vitro, and porcine ex vivo models, as it improves the mobility of skin cells, prevents the effect of lipopolysaccharides, and inhibits bacterial growth [66].

Kefir, a natural probiotic, has been associated with the enhanced proliferation and migration of human dermal fibroblast (HDF) cells in vivo, in a mouse model. In addition, a reduction in IL-1β and TGF-β1 expression, compared to the negative control, was observed in the kefir-administered animals. The angiogenesis, migration, and proliferation of fibroblasts were also enhanced by kefir administration, and improved fibrous connective tissue formation in the wound area was observed. These findings demonstrate that kefir is capable of accelerating the healing of burn wounds [68].

## 4. Human Studies Supporting the Role of Probiotics in Wound Healing

Recently, numerous studies and clinical trials have highlighted the impact and promising role of oral and topical probiotics in the prevention and treatment of various cutaneous diseases, including chronic wounds from burn patients, surgical lesions, psoriasis, dermatitis, acne vulgaris, or diabetic foot ulcer (DFU) [69].

As revealed by Twetman et al. (2018), topical and systemic administration of probiotics may stimulate the wound healing of oral mucosa through the modulation of matrix metalloproteinases production and interferons levels, in a randomized, placebo-controlled double-blind cross-over study comprising healthy volunteers. The subjects involved in this study were treated with lozenges which contained a mix of probiotic strains, such as *L. reuteri* DSM 17938 and ATCC PTA 5289, each with 5 × 10^9^ colony-forming units (CFU)/lozenge, for 8 days. They also received a topical application of the same probiotic in form of oils. However, the results of this placebo-controlled, randomized, double-blind clinical trial indicated that the limited number of participants (10 volunteers) and the large variations in the measured parameters represent a significant difficulty regarding the demonstration of the impact of oral and systemic probiotics on the healing process through immune modulation [70].

Additionally, several studies demonstrated that single-strain or multistrain probiotics may stimulate IgA and IgG synthesis and decrease IL-6 levels in burned patients. For instance, a clinical trial conducted at an intensive care and burn unit investigated the efficacy of single-strain probiotics that contained *S. thermophilus*, *L. acidophilus*, and *B. longum,* and that of mixed-strain probiotics including *Streptococcus salivarus subsp. Thermophilus*, *B. breve*, *B. longum*, *L. rhamnsosus*, *L. delbrueckii,* and *L. acidophilus,* in form of capsules, for 14 consecutive days. The authors concluded that the mixed-strain probiotics significantly reduced the inflammation marker IL-6 level, while the single-strain probiotics increased IgA production [71]. A similar study, performed by Perdanakusuma et al., investigated the effects of single-strain probiotics in the treatment of patients with thermal burns. The administration of oral probiotics including *Bifidobacterium infantis* 35624 or *Lactobacillus reuteri protectis* for 14 consecutive days indicated that although the *Bifidobacterium infantis* single-strain probiotic stimulates IgA production, for the treatment of thermal burns, supplementation with mixed-strain probiotics is recommended.

More recently, another study evaluated the efficacy of the oral administration of multistrain probiotics, selected based on previous investigations, composed of *L. rhamnosus* CECT 8361, *B. longum* CECT 7347, and *B. lactis* CECT 9145, in combination with corticosteroid and calcipotriol, on patients diagnosed with psoriasis. A total of 90 patients were selected for this clinical trial (45 patients were enrolled in the placebo group, and 45 patients were enrolled in the probiotic group). For 12 weeks, patients in the probiotic group received a capsule of probiotics in combination with topical application of a corticosteroid and calcipotriol, while patients in the placebo group received a capsule of maltodextrin. The results of this study showed that oral probiotics could reduce the severity of cutaneous diseases when they are administered with topical corticosteroid. Moreover, probiotic blends could induce beneficial effects on the evolution of psoriasis and on the composition of gut microbiota, according to Navarro-López [72].

Along with increasing evidence of the remarkable effects of probiotics described in several clinical trials, numerous studies summarized the usefulness of oral probiotic supplementation. This approach is gaining attention as a new therapeutic strategy for its ability to promote the healing process by preventing postoperative or surgical site infections, increasing colonic and gut motility, and alleviating postoperative complications [73]. A clinical controlled, randomized trial performed on 103 patients with multiple trauma who required surgical intervention revealed that the incidence of surgical site infections could be reduced through the oral administration of a combination of probiotics, including *Lactiplantibacillus plantarum* UBLP-40, *L. acidophilus* LA-5, *S. boulardii* Unique-28, and *Bifidobacterium animalis subsp. lactis* BB-12 with 0.5—1.75 × 1^9^ CFU/capsule.

The results of that study indicate that ingested probiotics exerts a positive impact on wound healing through immune modulation, and even on the prevention of pathogenic microorganisms’ invasion at the surgical site [74].

## 5. Wound Dressings with Probiotics and Prebiotics

The development of wound dressing technology has yielded promising findings due to its capacity to provide barriers and to promote healing through immune modulation and tissue regeneration of several (non-)infected wounds [75].

Dressings embedding probiotics and their derived compounds have been highly investigated in recent years. Numerous biopolymers and hydrogels have been recently investigated due to their unique properties, including biocompatibility, antibacterial effects, control of microbial attachment, colonization and biofilm development, biodegradability, low cytotoxicity, and even the ability to incorporate and increase the beneficial effect of probiotics and prebiotics in wound healing. These properties make them efficient candidates for wound dressings that can be used as adjuvant therapy.

Prebiotics, such as compounds of the carbohydrate group, often oligosaccharides, are able to stimulate an immune response due to their capacity to modulate the stability, composition, survival rate, and metabolic activity of probiotic cells. Known as synbiotics, combinations of probiotics and prebiotics, they have gained recognition for their effectiveness in the prevention of polymicrobial infections in the wound environment [76].

Currently, the most explored approaches for obtaining advanced wound dressings are represented by bioactive polymeric coatings and their derivatives, hydrogels, including collagen-based dressings, pectin, chitosan, or (sodium) alginate [77]. These materials can be easily obtained and combined with other active molecules to enhance their wound-healing efficacy. Novel drug delivery systems, including the co-encapsulation of active substances such as probiotics and prebiotics in topical formulations, represent a promising and innovative therapeutic approach for wound recovery and healing due to their synergistic effects, offering a new perspective for the development of the ideal wound dressing [77,78]. The main advantages of dressings containing probiotics and prebiotics are illustrated in Figure 2 below.

Alginate can be obtained from brown algae or extracted from *Pseudomonas* spp. bacteria. Thanks to their capacity to protect probiotic strains, alginate polymers can be used as efficient delivery systems. Notably, sodium alginate has gained recognition as a suitable natural coating polymer for encapsulating prebiotics and probiotics because of its effectiveness in wound healing, which is due to its biodegradable and non-toxic properties [79]. However, alginate has no intrinsic antibacterial properties; therefore, such dressings present the significant drawback of becoming contaminated with pathogenic bacteria, either colonizing the wound or present in the environment.

Pectin hydrogels encapsulated with probiotics are also able to enhance the therapeutic delivery to the targets, exhibiting strong antimicrobial and immunomodulatory effects [80].

Chitosan-based wound dressings exhibit excellent antibacterial, antifungal, and antiviral effects, as well as biodegradability and biocompatibility features, being applied to wound healing due to their unique properties. A well-investigated system for the encapsulation of probiotic strains and prebiotics is represented by alginate–chitosan compounds, for their ability to enhance probiotic colonization, but also to reduce the growth and development of wound pathogens, due to the intrinsic antibacterial properties of chitosan [77].

Current trends attempt to design new coating systems based on polymers and topical formulations, with incorporated probiotics and prebiotics, which could exhibit antibacterial properties and enhance the potential for accelerated wound healing [59].

Microencapsulation encloses bioactive molecules such as antimicrobials, enzymes, nutrients, drugs, vitamins, probiotics, etc., with polymeric materials. In the case of probiotics, microencapsulation can be conducted through various chemical and physical methods, which increase bioavailability and antimicrobial activity. Therefore, the survival rate of probiotics depends on various factors, including culture strain, particle size, conditions, or the nature of the encapsulating polymers [77].

Moreover, current research, which encompasses both in vivo and in vitro studies, has provided evidence that wound dressings incorporating probiotics and prebiotics have sparked interest due to their antibacterial activities and promising role in immunomodulation.

Biocompatible polymeric coatings containing combinations of probiotics, particularly *Lactobacillus* strains (*L. plantarum*, *L. brevis*, *L. plantarum*, *L. paracasei),* and prebiotics may accelerate wound healing, according to the latest research in the wound management area. Some of the findings are highlighted in Table 3.

As an example, a study conducted by Farahani and collaborators aimed to evaluate the therapeutic effects of encapsulated probiotics (*L. plantarum*) and prebiotics (fructooligosaccharide) in a mixture of polymers, including chitosan, pectin, sodium alginate, and calcium chloride. In vitro assessments were performed to optimize the microencapsulated polymers and investigate the effectiveness against pathogen agents such as *P. aeruginosa* ATCC 9027 and *S. aureus* ATCC 6538.

In vitro experiments were performed on 48 Wistar rats with induced, infected burn wounds. The wound dressings, which contained *L. plantarum* and prebiotics, were applied daily for 21 consecutive days.

The findings from their study revealed the positive impact of probiotic- and prebiotic-loaded particles on the modulation of wound inflammatory responses, through the comparison of microbial, histological, and clinical wound status. The results demonstrated that these formulations significantly improved antibacterial efficacy, possessed appropriate properties for healing infected wounds, and showed significant potential for use as curative and preventive treatment of burn wounds [81].

A recent study focused on developing wound dressings based on bacterial hydrogels encapsulating the probiotic strain *Lacticaseibacillus paracasei* TYM202 and extracellular polysaccharides 9EPS0 from *Bacillus velezensis* M76T11B, as a prebiotic. The authors evaluated the dressing’s efficiency in a rat wound model. The rats were divided into four groups (12 rats per group), and their wounds were covered with different types of treatments, which were changed daily: saline solution, hyaluronic acid (HA), extracellular polysaccharides 9EPS0 from *B. velezensis* M76T11B (HAEPS), and hyaluronic acid encapsulated with *L. paracasei* TYM202.

The results revealed that probiotic and prebiotic hydrogels possess favorable mechanical properties and exhibit antimicrobial behavior against Gram-negative (*E. coli*) and the Gram-positive model strains (*S. aureus*), maintaining wound homeostasis by providing favorable conditions for the skin microbiota and offering protection from pathogenic agents. The prebiotic compounds (extracellular polysaccharides 9EPS0 from *B. velezensis* M76T11B) promoted the growth and metabolism of the probiotic strain incorporated into the hydrogel (*L. paracasei* TYM202), effectively maintaining the activity of the probiotic strain. Along with their antibacterial impact, probiotic hydrogels possess a remarkable ability to promote wound healing, due to immune regulation. In this context, this innovative approach can regulate collagen deposition, stimulate angiogenesis, and reduce inflammation in a rat model [82].

**Table 3 ijms-25-05723-t003:** In vivo and in vitro studies highlighting the novel wound dressing with probiotics and prebiotics, with potential applications in wound healing.

Type of Study	Dressing Type	Probiotics/ Prebiotics	Pathogenic Indicator Microorganisms	Effects/Mode of Action	Intended Application	References
in vivo (mice model) in vitro	Sponge dressings	*L. plantarum* UBLP-40 (MTCC 5380)	*S. aureus* 9144	Decrease bacterial attachment and growth; Promote tissue granulation. Accelerate wound healing; Decrease in matrix metalloproteases, TNF-α levels. Increase in TGF-β, VEGF, antioxidant enzymes levels.	Improved treatment of chronic infected wounds	Sandhu et al., 2023 [83]
in vivo (rat model)	Hiybrid bilayer wound dressing	*L. brevis* (KCTC 3498)	*S. aureus* subsp. *Aureus* KCCM 40050	Growth inhibition in planktonic and biofilm bacterial cultures; Reduction in the number of mast cells, microvessels, granulation tissue area.	Better recovery and treatment of infected wounds	Kim et al., 2022 [78]
in vivo (rat model)	Probiotic hydrogels	*Lacticaseibacillus paracasei* (TYM202), extracellular polysaccharides 9EPS0 from *B. velezensis* (M76T11B)	*E. coli*, *S. aureus*	Inhibition of growth and attachment of the pathogenic agents; Reduced inflammation; Stimulated collagen deposition; Promoted angiogenesis.	Improved treatment of skin conditions	Xu et al., 2024 [82]
in vivo (Wistar rat model) in vitro	Microencapsulated probiotic cells and prebiotics in different types of coatings (pectin, sodium alginate, and chitosan)	*Lactiplantibacillus plantarum* (ATCC 1058), fructooligosaccharide (FOS)	*P. aeruginosa* ATCC 9027, *S. aureus* ATCC 6538	Accelerate wound healing process by increasing wound closure rate; Inhibition of growth and biofilm formation in the evaluated Gram-negative strains.	Improved treatment of infected burn wound	Farahani et al., 2023 [81]

Such wound dressings with incorporated probiotics and prebiotics have the potential to prevent and to treat resistant wound infections in both acute and chronic patients.

As most studies on probiotics and probiotic coating technology have been predominantly performed on animal models, mostly rodents, it is essential to expand their investigation to more clinical trials, in order to investigate key aspects regarding the potential benefits of this approach in the wound healing process for various type of patients.

## 6. Conclusions

Probiotics could be highly used in the prevention and adjuvant treatment of a wide range of wounds that are usually complicated by microbial infections, including surgical wound infections, burn wounds, oral mucosa injuries, or diabetic foot ulcers. Numerous in vitro and in vivo studies performed on tissue cultures, animals, and clinical trials have shed light on the beneficial effects and mechanisms of oral and topical probiotic formulations in addressing wound healing. Wound dressings and coatings embedding probiotics and/or prebiotics are currently under intensive research, as they could improve the management of difficult-to-treat wounds.

Dressings containing probiotic and prebiotic components offer multiple advances in healing and on-site infection control. Researchers are also investigating approaches to reduce their drawbacks, such as the potential of contamination, and to tailor their intrinsic antibacterial properties, in order to design optimal solutions for particular types of wounds. The design of such materials currently represents a priority research focus, and we expect an increasing number of studies and clinical trials aimed at their evaluation in the next few years.

## Figures and Tables

**Figure 1 ijms-25-05723-f001:**
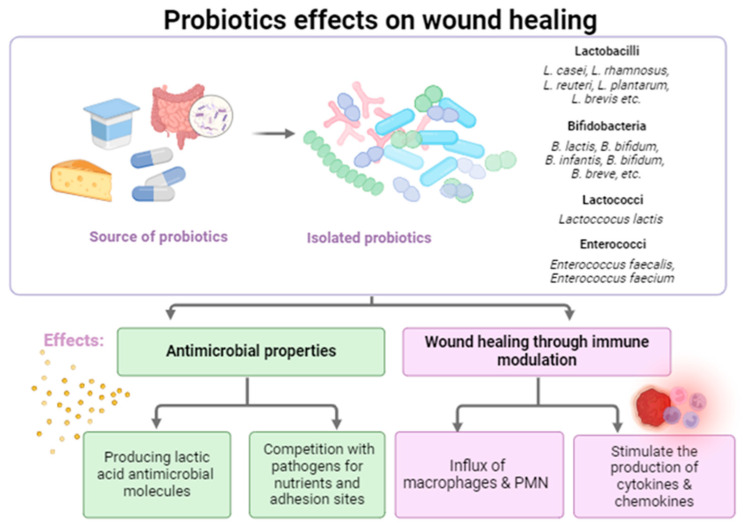
Probiotic mechanisms associated with improved wound healing (created with BioRender.com, accessed on 31 March 2024).

**Figure 2 ijms-25-05723-f002:**
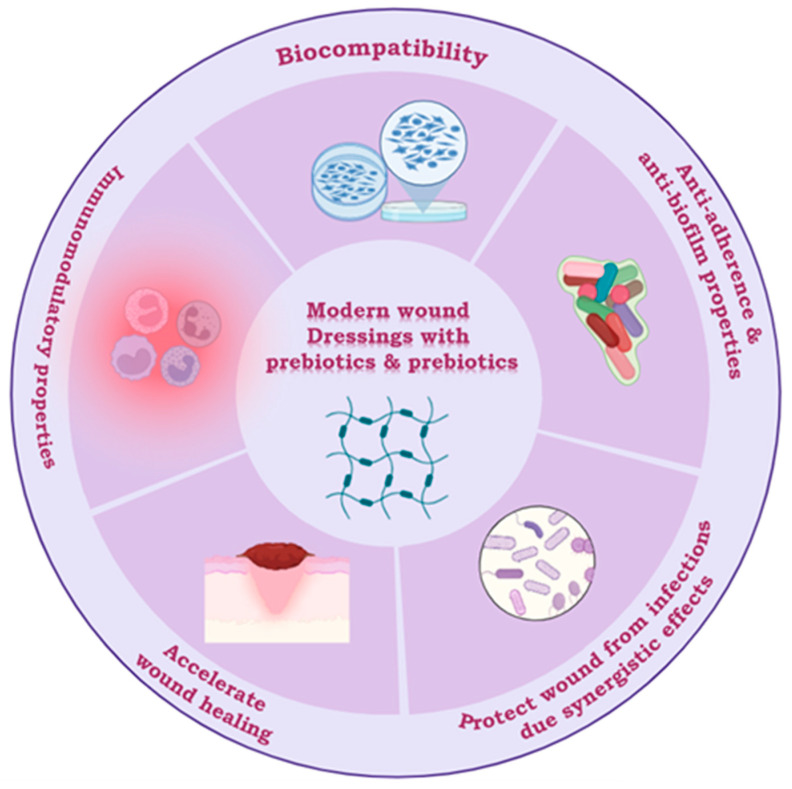
Properties and main advantages of probiotic- and prebiotic-containing dressings (created with BioRender.com, accessed on 10 April 2024).

**Table 1 ijms-25-05723-t001:** In vivo and clinical studies investigating the impact of oral probiotics in wound healing.

In Vivo Model/Clinical Trial	Probiotics Used	Method of Administration	Effects Observed in Wound Healing	Intended Application	References
Swiss mice with excisional skin wounds	*L. johnsonii* LA1, *L. paracasei* ST11, and *L. rhamnosus* LPR	Oral gavage	Oral gavage administration of *L. rhamnosus* LPR enhanced wound healing by promotion of reepithelization process and angiogenesis, decreased leukocyte infiltration and granulation tissue.	Oral treatment with probiotics for stimulating wound healing	Moreira et al., 2021 [39]
Wistar rats with experimentally induced wounds	Probiotic mix: *L. paracasei* 37, *L. rhamnsus* HN001, *B. lactis* HN0019, *L. acidophilus* NCFM	Oral administration of probiotics (dose: 250 mg/day).	The reduced expression of IL-17, IL-6, TNF-α (Tumour Necrosis Factor α), and TGF-β (Transforming Growth Factor β) improved wound healing and induced faster recovery. The decreased inflammatory process accelerated the deposition of collagen and fibrosis phase.	Oral treatment with probiotics for stimulating wound healing	Tagliari et al., 2022 [29]
Clinical trial: females with episiotomy	Capsules of *Lactobacillus casei* 431 (once a day, for two weeks)	Oral administration of probiotic capsules	Orally ingested probiotics stimulated the wound healing process through angiogenic effects and anti-inflammatory properties.	Oral supplementation of probiotics as adjuvants in wound healing	Abdollahpour et al., 2023 [33]
Clinical trial: burn patients	*L. rhamnosus*, *L. delbrueckii subsp. Bulgaricus*, *L. acidophilus*, *L. casei*, *Streptococcus salivarus subsp. Thermophilus*, *B. breve*, *B. longum*	Oral supplementation in form of capsules	Promoted immune response by reducing neutrophil and leucocyte levels; antimicrobial properties against burn wound pathogens; improved wound healing process.	Orally ingested probiotics for improving burn wound healing	Putra et al., 2017 [37]
Clinical trial: pediatric patients with surgical interventions	*L. rhamnosus* GG ATCC 53103	Oral-route administration as drops	Reduced the duration of antibiotic-associated diarrhea, frequency of dressing change, and incidence of postoperative wound complications.	Oral probiotics used as complementary therapy for management of complications which may occur after surgical intervention.	Esposito et al., 2018 [36]
Clinical trial: oral mucosa injuries	*L. reuteri* (DSM 17938 and ATCC PTA 5289)	Oral administration of probiotics in form of lozenges	Alleviated clinical symptoms, less tissue swelling.	Alternative therapy with oral probiotics	Wälivaara et al., 2019 [34]
Clinical trial: diabetic foot ulcer (DFU)	*L. fermentum*, *L. acidophilus*, *L.casei*, *B. bifidum*	Oral administration of probiotics mix incorporated in capsules	Probiotics reduced the ulcer length and improved parameters associated with healing process.	Oral supplementation for symptom alleviation in DFU patients	Mohseni et al., 2018 [35]

**Table 2 ijms-25-05723-t002:** In vivo, ex vivo, and in vitro studies highlighting the impact of topical probiotics in wound healing.

Type of Study	Efficient against Wound Pathogen	Probiotics Used	Method of Evaluation	Effects	Intended Application	References
In vitro study	*S. aureus*	Lysate or spent culture fluid of *L. rhamnosus* GG, ATCC 53103, *L. reuteri* (ATCC 55730), and *L. salivarius* (UCC118)	Co-culture assays of normal human epidermal keratinocytes (NHEK).	Stimulate bacteriocins production, which enhances antimicrobial effects.	Topical therapy for wound infections.	Mohammedsaed et al., 2014 [46]
In vitro study	*P. aeruginosa* isolated from burn and wound infections	*L. rhamnosus* *GG*, *L. acidophilus*	Well diffusion method.	Enhanced antimicrobial effects of bacteriocins produced by probiotics.	Prevention and treatment of wound infections.	Al-Malkey et al., 2017 [47]
In vitro study	*E. coli*, *P. aeruginosa*, *Propionibacterium acnes*	Supernatants of *L. delbrueckii*, *L. brevis* D-24, *L. acidophilus* La-5, L-10, L-26, *L. plantarum* 226v, *L. casei* 20021 *L.salivarius* 20555, *B. animalis* Bb12, *B. lactis* B-94, *B. longum* 20088	Well diffusion method.	Antibiofilm effects against tested pathogens; inhibition of bacterial adherence.	Topical application to restore cutaneous dysbiosis.	Lopes et al., 2017 [48]
In vitro study	*Enterobacter hormaechei*, *K. pneumoniae*, *A. baumannii*	*L. reuteri* SD2112	Co-culture with human diploid cells.	Inhibition of bacterial adherence.	Topical application for improving the management of open wounds and burns.	Chan et al., 2018 [49]
In vitro study	*P. aeruginosa* *S. aureus*	*L. acidophilus* CL1285, *L. casei* LBC80R, *L. rhamnosus* CLR2	Co-culture assays. Encapsulation of probiotics.	The encapsulation of probiotics in combination with antibiotics promoted complete eradication of tested pathogens	Topical co-administration with antibiotics, to increase their efficiency.	Li et al., 2018 [50]
In vitro study	*P. aeruginosa*	*L. reuteri* DSM17938, *L. acidophilus* DSM, *Bacillus coagulans*, *L. plantarum* 299v, *B. bifidum* DSM20456	Well diffusion method.	Some combinations of probiotics and antibiotics manifested synergistic effects.	Treatment of wounds through combination of probiotics and antibiotics for a better outcome.	Moghadam et al., 2018 [51]
In vitro study	*P. aeruginosa*, *S. aureus* MRSA (Methicillin-resistant *S. aureus*)	Supernatant solution of *L. plantarum* F10	Well diffusion method. Co-culture assays.	Biofilm inhibition effects.	Improving the treatment of cutaneous infections.	Onbas et al., 2019 [52]
1. Skin from dorsal part of domestic farm pig (ex vivo). 2. Rat wound model (in vivo)	*S. aureus*	Protein-rich fraction of *L. plantarum* USM8613 isolated from meat products	Topically treated wound infections inoculated with *S. aureus*.	Promoted wound healing by stimulating the synthesis of chemokines and cytokines such as TNF—γ, INF—α, IL—4, IL—6, and β-defensin expression through all the healing stages. Stimulated keratinocytes migration rate. Reduced *S. aureus* infection at wound sites through autolysis process.	Topically treatment of probiotics in infected wounds.	Ong et al., 2019 [30]
Immunocompromised wound mice	*A. baumanii* MDR (Multidrug resistant) clinical isolates	Supernatant of *L. acidophilus* ATCC 4356, *L. casei* ATCC 393, *L. reuteri* ATCC 23272	Topically treatment of *A. baumanii* infected wounds.	Enhanced wound closure. Growth inhibition of planktonic cells.	Topical therapy for enhancing wound closure.	Stanbro et al., 2019 [43]
1. In vitro study. 2. In vivo Wistar rats wound model	*S. aureus* MTCC No-3160	Probiotic gel formulation and supernatant solution of *Lactobacillus* VITSAMJ1 (isolated from goat milk)	Agar-well diffusion method. Topically treated infections with probiotic gel formulation (twice a day).	Prevention of wound infection through synthesis of antimicrobial substances, which inhibit bacterial adhesion to epithelial tissue. The probiotic gel formulation reduced the size of the wound, accelerated the healing process by promoting angiogenesis and inflammatory responses.	Development of a topical product for enhanced wound healing.	Sinha et al., 2019 [44]
1. In vitro study. 2. In vivo Wistar rats wound model	*P. aeruginosa* clinically isolated from burn wounds and *P.aeruginosa* ATCC 27853	Supernatant of *Bacillus coagulans* (DSM1), *B. bifidum* (DSM20456) and *L. plantarum* 299v (DSM9843), *L. salivarius* ES1, *L. reuteri* ES10 and *L. salivarius* ES8	Disk diffusion method (NCCLS) Topical wound treatment.	Inhibition of *P. aeruginosa* growth. Promoted wound closure and significantly reduced wound size.	Improved topical wound treatment.	Moghadam et al., 2020 [53]
In vivo Wistar rats wound model	*P. aeruginosa* MDR clinical isolates from patients with burn wounds	Supernatant of *L. casei* PTCC 1608—spray	Topical treatment (wounds were sprayed every day for 28 days).	Anti-adhesion effects on tested bacterial strains. Fibrogenesis process stimulation.	Improved topical therapy for wound infections caused by MDR *P. aeruginosa.*	Abootaleb et al., 2021 [42]
In vitro study	*P. aeruginosa* strains clinically isolated from patients with wound infections	*L. plantarum*	In vitro treatment of pathogen agents with probiotics.	Modulation of *P. aeruginosa* virulence factor production (elastase and pyocyanin).	Improved topical therapy of *P. aeruginosa* wound infections.	Shams Eldeen et al., 2021 [54]
1. In vitro study. 2. Wound models in Wistar rats	*E. coli*, *S. aureus*, and *Salmonella* spp.	*L. reuteri* encapsulated in hydrogels	Well diffusion method. Co-culture assays. Topically treated wound infections in rats.	Antibacterial activity in vitro and in vivo. Antibiofilm properties in vivo. Stimulated wound healing, promoted collagen deposition (in vivo).	Hydrogels dressings for topically treatment of infected wounds	Ming et al., 2021 [41]
Ex vivo skin model (culture of human dermal fibroblasts, HDFs)	*P. aeruginosa* biofilms	Capsules containing *L. acidophilus* CL128, *L. rhamnosus* CLR2, and *L. casei* LBC80R	In vitro biofilm assay Co-culture (probiotics—HDFs)	Probiotics treatment stimulated cell migration and eradicated preformed biofilms (in vitro), decreased the bioburden in wound infection (ex vivo).	Direct application of probiotics and probiotic wound dressings for improving wound healing.	Li et al., 2023 [55]
Clinical Trials (patients with diabetic foot ulcer)	*Streptococcus* sp., *Stpahylococcus* sp., *P. aeruginosa* biofilms	*L. plantarum* ATCC 10241	Topical application on the wounds.	Probiotics improved angiogenesis in the skin tissues of patients with diabetes mellitus and diabetic foot ulcer, while reducing microbial load on site.	Adjuvant to surgical debridement to accelerate healing process.	Argañaraz Aybar et al., 2022 [56]

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
