# Peer review of "Probiotics in Wound Healing"

_ijms, 2024, doi:10.3390/ijms25115723_

Round 1

Reviewer 1 Report

Comments and Suggestions for Authors

While I feel the topic is worthwhile, there are many flaws with this paper:

1) You present many papers as though they are totally supportive of probiotics, however, there are many potential flaws in the studies that should discussed. Many studies have no controls, or not even related to wound healing (a dermatitis study). For instance, the hypospadias study does not have a probiotic-only control. I feel the paper would be improved if you pointed out the weaknesses of the studies, because there are many flaws.

2) I would also not say that probiotics have "antimicrobial" activities. They may prevent bacterial infections, but I see no evidence that they can treat active infections. In addition, I agree that probiotics may reduce bacterial-induced diarrhea, but that does not mean they improve wound healing. 

3) There were many studies in the 1970s and 1980s that showed that low dose pathogenic bacteria (Staph, Strept, Gram negatives, etc.) would stimulate healing by increasing inflammation. Many bacterial byproducts also improved healing in the same manor. Probiotics probably do the same thing. It might also be worthwhile to consider comparing probiotic wound dressings with no antimicrobial (ie. sterile). 

4) I do not see how probiotics directly affect the proliferative phase of wound healing. At that stage, the epithelium should have closed the wound. If the wound is still open, the inflammation persists, and the wound still is influenced by the inflammatory phase. 

5) Wound dressings, such as alginates, have been used for decades, and while you could assume that they enhance probiotic bacteria, they do become infected. The evidence that they promote probiotic organisms is quite weak. 

6) You have problems with the references. First of all, you have 61 references in the text, but list only 33 references. In addition, reference 12 states the year of publication as 1912. I doubt that is the correct date. 

Comments on the Quality of English Language

There are many problems with the grammar and spelling. There are multiple changes in tense and many words are misspelled or have hyphens (-) in the middle of them. The paper should be reviewed by someone with expertise in the English language. 

Author Response

Dear Reviewer 

Many thanks for your useful recommendation and corrections. They helped us to
significantly improve our paper.
Please find bellow the point-by-point reply to the reviewers’ comments:

“While I feel the topic is worthwhile, there are many flaws with this paper:
1) You present many papers as though they are totally supportive of probiotics, however,
there are many potential flaws in the studies that should discussed. Many studies have
no controls, or not even related to wound healing (a dermatitis study). For instance,
the hypospadias study does not have a probiotic-only control. I feel the paper would
be improved if you pointed out the weaknesses of the studies, because there are many
flaws.”
Answer: Dear reviewer, we are very grateful for your corrections and recommendations.
Considering your comments, we have made significant changes in our paper, starting with the
introduction. Among the corrections, we have removed the information that was not related
to wound healing (i.e dermatitis study), and also highlighted some important weaknesses of
the discussed studies, empathising on what should have been done differently in the
respective research to increase the scientific knowledge on the field.
2) “I would also not say that probiotics have "antimicrobial" activities. They may
prevent bacterial infections, but I see no evidence that they can treat active infections.
In addition, I agree that probiotics may reduce bacterial-induced diarrhea, but that
does not mean they improve wound healing. “
Answer: We agree with your comment and we have rephrased the text accordingly. You will
find that we have referred to the anti-pathogenic properties of probiotics, and offered concrete
examples on what anti-pathogenic mechanisms could be potentially related to the prevention
of wound infection. Moreover, we referred to a recent study highlighting that the application
of probiotics could induce different growth-reducing capacities towards relevant wound
pathogens in a novel in vivo like human plasma biofilm model (hpBIOM). This study showed
that B. lactis and S. cerevisiae possess slight bacteria-reducing properties, while the survival
of the opportunistic yeast C. albicans was not affected at all. The most significant
antimicrobial activity was detected after the treatment with L. plantarum in all the evaluated
wound pathogens. [Besser M, Terberger J, Weber L, Ghebremedhin B, Naumova EA, Arnold
WH, Stuermer EK. Impact of probiotics on pathogen survival in an innovative human plasma
biofilm model (hpBIOM). J Transl Med. 2019 Jul 25;17(1):243. doi: 10.1186/s12967-019-
1990-4.]
3) “ There were many studies in the 1970s and 1980s that showed that low dose
pathogenic bacteria (Staph, Strept, Gram negatives, etc.) would stimulate healing by
increasing inflammation. Many bacterial byproducts also improved healing in the

same manor. Probiotics probably do the same thing. It might also be worthwhile to
consider comparing probiotic wound dressings with no antimicrobial (ie. sterile). “
Answer: Most of the discussed studies regarding the efficiency of probiotic/prebiotic wound
dressings contain controls (i.e. no probiotic control, dressing control). We have tried to
consider the most relevant and recent papers for our study, in order to offer an updated
overview regarding the current trends in wound healing, with the help of probiotics.
4) “I do not see how probiotics directly affect the proliferative phase of wound healing.
At that stage, the epithelium should have closed the wound. If the wound is still open,
the inflammation persists, and the wound still is influenced by the inflammatory
phase. “
Answer: We have massively reformulated the respective section, in order to better understand
the proposed mechanisms of probiotics in wound healing. Thank you for your
recommendation.
5) “Wound dressings, such as alginates, have been used for decades, and while you could
assume that they enhance probiotic bacteria, they do become infected. The evidence
that they promote probiotic organisms is quite weak. “
Answer: Thanks for spotting out this drawback. We have now updated the information in the
respective section and commented accordingly.
“6) You have problems with the references. First of all, you have 61 references in the text, but
list only 33 references. In addition, reference 12 states the year of publication as 1912. I
doubt that is the correct date. “
Answer: We have revised references. Apologies for this mistake in the previous paper draft!

“Comments on the Quality of English Language
There are many problems with the grammar and spelling. There are multiple changes in tense
and many words are misspelled or have hyphens (-) in the middle of them. The paper should
be reviewed by someone with expertise in the English language.”
Answer: We have significantly revised our paper and corrected language mistakes and hypos.

Reviewer 2 Report

Comments and Suggestions for Authors

The review deals with a very interesting and innovative topic so the hope is to be able to accept it for publication! The authors rightly focus on the use of probiotics since they, according to Scientific Association of Probiotics and Prebiotics, are "live microorganisms which provide health benefits when they are consumed in an adequate manner". In fact Antibiotic-resistant microorganisms represent a serious global concern, whereas they increase the risk of developing healthcare-associated infections (HAI) in hospital and surgical units Inadequate and overuse of antibiotics has led to antibiotic-resistant infections caused by resistant microorganisms which are a major cause of chronically infected wounds. Furthermore, tolerant microorganisms causing biofilm infections are another huge threat in the development of chronic and difficult-to-treat wounds, more efficient approaches to fight such microbial challenges are necessary and represent a global priority.

The limitation of this review is that the authors focus precisely on this topic, limiting the problems relating to acute and chronic wounds. Better, and this for reasons of order, it would be better to initially dedicate space to the problems relating to the healing of wounds, introducing the potential reader to this problem and not simply considering it as an obvious fact. Therefore the review should begin with paragraphs relating to the mechanisms of wound healing considering how these mechanisms are different from each other. Only by introducing these issues will we have a complete vision of the topic addressed. When describing these mechanisms, authors should be careful to focus only on recent articles (2021-2024) as the focus on wound healing has significantly increased judging by the literature.

Comments on the Quality of English Language

Moderate editing of English language required

Author Response

Dear Reviewer

Many thanks for your useful recommendation and corrections. They helped us to significantly improve our paper.

Please find bellow the point-by-point reply to the reviewers’ comments:

“The review deals with a very interesting and innovative topic so the hope is to be able to accept it for publication! The authors rightly focus on the use of probiotics since they, according to Scientific Association of Probiotics and Prebiotics, are "live microorganisms which provide health benefits when they are consumed in an adequate manner". In fact Antibiotic-resistant microorganisms represent a serious global concern, whereas they increase the risk of developing healthcare-associated infections (HAI) in hospital and surgical units Inadequate and overuse of antibiotics has led to antibiotic-resistant infections caused by resistant microorganisms which are a major cause of chronically infected wounds. Furthermore, tolerant microorganisms causing biofilm infections are another huge threat in the development of chronic and difficult-to-treat wounds, more efficient approaches to fight such microbial challenges are necessary and represent a global priority.

The limitation of this review is that the authors focus precisely on this topic, limiting the problems relating to acute and chronic wounds. Better, and this for reasons of order, it would be better to initially dedicate space to the problems relating to the healing of wounds, introducing the potential reader to this problem and not simply considering it as an obvious fact. Therefore the review should begin with paragraphs relating to the mechanisms of wound healing considering how these mechanisms are different from each other. Only by introducing these issues will we have a complete vision of the topic addressed. When describing these mechanisms, authors should be careful to focus only on recent articles (2021-2024) as the focus on wound healing has significantly increased judging by the literature.”

Answer: Dear reviewer, thank you very much for your appreciation and valuable comments. We have massively revised our paper, accordingly, and included paragraphs related to the general wound healing mechanisms as suggested. Moreover, in the “Mechanisms of probiotics in wound healing” section we have commented how probiotics could modulated these mechanisms and promote wound healing. Recent examples were also offered in the section “4. Human studies supporting the role of probiotics in wound healing”.

“Comments on the Quality of English Language

Moderate editing of English language required”

Answer: We have massively revised our paper and corrected language mistakes and hypos.

Thank you for your valuable time spent in revising our paper.

Alina-Maria Holban

Round 2

Reviewer 1 Report

Comments and Suggestions for Authors

The authors have greatly improved the paper and have addressed my concerns. 

Reviewer 2 Report

Comments and Suggestions for Authors

The authors have asked correctly to my questions

Comments on the Quality of English Language

Moderate editing of English language required